# Proteomics Analysis Reveals That Caspase-Like and Metacaspase-Like Activities Are Dispensable for Activation of Proteases Involved in Early Response to Biotic Stress in *Triticum aestivum* L.

**DOI:** 10.3390/ijms19123991

**Published:** 2018-12-11

**Authors:** Anastasia V. Balakireva, Andrei A. Deviatkin, Victor G. Zgoda, Maxim I. Kartashov, Natalia S. Zhemchuzhina, Vitaly G. Dzhavakhiya, Andrey V. Golovin, Andrey A. Zamyatnin

**Affiliations:** 1Sechenov First Moscow State Medical University, Institute of Molecular Medicine, Trubetskaya str., 8, bld. 2, Moscow 119991, Russia; balakireva.anastacia@gmail.com (A.V.B.); andreideviatkin@gmail.com (A.A.D.); golovin.andrey@gmail.com (A.V.G.); 2Institute of Biomedical Chemistry, Pogodinskaya str., 10, bld. 8, Moscow 119121, Russia; victor.zgoda@gmail.com; 3All Russian Research Institute of Phytopathology, VNIIF, Bolshie Vyazemi, Odintsovsky distr., Moscow region 143050, Russia; maki505@mail.ru (M.I.K.); zhemch@mail.ru (N.S.Z.); dzhavakhiya@yahoo.com (V.G.D.); 4Faculty of Bioengineering and Bioinformatics, Moscow State University, Moscow 119992, Russia; 5Belozersky Institute of Physico-Chemical Biology, Lomonosov Moscow State University, Moscow 119992, Russia

**Keywords:** degradome, wheat, cultivar, protease, papain-like cysteine protease (PLCP), subtilase, metacaspase, caspase-like, wheat leaf rust, *Puccinia recondita*, *Stagonospora nodorum*

## Abstract

Plants, including *Triticum aestivum* L., are constantly attacked by various pathogens which induce immune responses. Immune processes in plants are tightly regulated by proteases from different families within their degradome. In this study, a wheat degradome was characterized. Using profile hidden Markov model (HMMer) algorithm and Pfam database, comprehensive analysis of the *T. aestivum* genome revealed a large number of proteases (1544 in total) belonging to the five major protease families: serine, cysteine, threonine, aspartic, and metallo-proteases. Mass-spectrometry analysis revealed a 30% difference between degradomes of distinct wheat cultivars (Khakasskaya and Darya), and infection by biotrophic (*Puccinia recondita* Rob. ex Desm f. sp. tritici) or necrotrophic (*Stagonospora nodorum*) pathogens induced drastic changes in the presence of proteolytic enzymes. This study shows that an early immune response to biotic stress is associated with the same core of proteases from the C1, C48, C65, M24, M41, S10, S9, S8, and A1 families. Further liquid chromatography-mass spectrometry (LC-MS) analysis of the detected protease-derived peptides revealed that infection by both pathogens enhances overall proteolytic activity in wheat cells and leads to activation of proteolytic cascades. Moreover, sites of proteolysis were identified within the proteases, which probably represent targets of autocatalytic activation, or hydrolysis by another protease within the proteolytic cascades. Although predicted substrates of metacaspase-like and caspase-like proteases were similar in biotrophic and necrotrophic infections, proteolytic activation of proteases was not found to be associated with metacaspase-like and caspase-like activities. These findings indicate that the response of *T. aestivum* to biotic stress is regulated by unique mechanisms.

## 1. Introduction

Wheat (*Triticum aestivum* L.) is a major grain species of value to both industry and biotechnology. One of the main factors that influence the use of wheat is its resistance to pathogens. Resistance itself is defined by the fact that wheat is a hexaploid organism, which harbors three genomes with an overall haploid size of more than 15 Gbp [1]. Genome A was obtained from *Triticum urartu*, Genome B from an unknown grass related to *Aegilops speltoides* and Genome D from *Aegilops tauschii*. In addition to the complexity of wheat genomes, there are also variations between wheat cultivars. Differences in protein levels expressed amongst the cultivars have previously been analyzed and assessed as being up to 30% [2,3]. These differences determine wheat resistance to pathogens and growth conditions, and the suitability of wheat cultivars for different applications.

In fact, wheat constantly suffers from various pathogens: bacterial (*Pseudomonas* spp. [4], *Xanthomonas translucens* [5], etc.), fungal (*Puccinia recondita*, *Fusarium* spp., *Blumeria graminis*, *Zymoseptoria tritici* [6], etc.), viral (barley stripe mosaic virus [7], wheat streak mosaic virus [8], yellow leaf mosaic virus [9], etc.), herbivorous insects (*Sitobion avenae* [10]) and even nematodes (*Heterodera avenae* [11]). Pathogens have distinct strategies during plant cell infection: necrotrophic pathogens (*Botrytis cinerea* [12]) and herbivores [13] promote plant growth and lead to necrosis of the infected cell through consumption of its content, whilst biotrophic pathogens (*Pseudomonas syringae* [14]) feed on living cells, suppressing plant growth and launching programmed cell death (PCD) of infected cells.

PCD is regulated by proteolytic enzymes in living organisms and the degradome is an overall complex consisting of all the proteases in a cell. It is known that the human genome contains 588 genes for proteases, including about 150 proteases that are transcriptionally active, depending on the type of tissue [15]. The number of encoded proteases amounts to 723 in *Arabidopsis thaliana* L. [16], 997 for rice [17,18,19] and 901 for tomatoes [20].

The most widely studied form of PCD (apoptosis in humans) is regulated by proteolytic cascades, which include caspases—Asp-specific cysteine proteases. Caspases are divided into upstream initiator caspases (-2, -8, -9 and -10), and downstream executioner caspases (-3, -6 and -7) [21]. Caspases are synthesized as inactive procaspases. Upstream procaspases undergo autocatalytic processing and activate downstream procaspases through limited proteolysis. Downstream caspases activate proapoptotic proteins, including other proteases, through the proteolytic cascades that lead to massive degradation of proteins and cell death.

As with humans, plant PCD is also regulated through proteolytic cascades [21]. Although caspases are absent in plants, caspase-like activity can be detected in vegetation after induction of PCD with DEVDase [22,23], VEIDase [24] and YVADase [25] activities. These are attributed to proteases from different families, e.g., subtilases (S8) [26], proteasome subunits (T1) [23] and vacuolar processing enzymes (VPEs, C13) [27]. As with caspases, these proteases are synthesized as zymogens or preproenzymes and require proteolytic activation. They usually contain three regions: a signal peptide, an inhibitory prodomain and a proteolytic domain. A prodomain is often (autocatalytically) processed, resulting in the release of an active, mature enzyme. The other main characteristic of PCD in plants is metacaspase activity. Metacaspases cleave after R and K, and have been shown to participate in a variety of processes associated with the death of cells [28,29,30,31,32]. The nature of such processes in metacaspases has been described earlier [33]. It is worth noting that although a number of substrates are known for having caspase-like proteases and metacaspases [34], no data exist on their exact substrate specificity and the cellular proteolytic cascades with which they are associated.

In the case of wheat, its response to pathogens, e.g., to *Blumeria graminis* f. sp. *tritici* [35], or *Fusarium graminearum* [36], and to numerous abiotic stresses [37], has been characterized only in general terms without focusing on particular groups of proteins. Moreover, the degradome of wheat has not yet been characterized despite its importance for immunity and PCD-related processes. It is known that expression of the wheat metacaspase 1 gene (TaMCA1) increases when infected with *Puccinia striiformis* [38]. However, there are no data available on proteolytic cascades and proteases that regulate wheat immunity in healthy and infected plants.

Liquid chromatography-mass spectrometry (LC-MS) is a powerful technique, which has been successfully used to study the properties of different wheat cultivars [2,3,39], plant development [40,41], and wheat responses to biotic [35,36] and abiotic stresses [37]. An LC-MS approach has been adopted in this study for characterization of all wheat proteases that were identified using Pfam identificators attributed to all the proteases with the use of profile hidden Markov model (HMMER) algorithm [42] that was previously successfully used for identification of proteins [43]. We performed classification of identified proteases. Comparison of two wheat cultivars in terms of proteolytic activity was done. LC-MS enabled identification of changes in the wheat degradome upon infection by biotrophic (*Puccinia recondita* Rob. ex Desm f. sp. tritici) and necrotrophic (*Stagonospora nodorum*) pathogens, and assessment of the impact of these infections on proteolytic activity in wheat cells. LC-MS also facilitated analysis of proteases present in healthy and infected wheat plants, determining the role of caspase-like and metacaspase-like proteases in proteolytic cascades, and the overall proteolytic activity in wheat cells.

## 2. Results

### 2.1. Degradome of T. aestivum Is Represented by Diverse Protease Families

First, proteases encoded in the wheat genome were identified. Release 39 proteome of *T. aestivum* was obtained from the Ensembl genomes database (ftp://ftp.ensemblgenomes.org/) and used in identification of the wheat degradome. Protein families included cysteine, serine, aspartic, threonine, and metallo-proteases. The HMMER algorithm [42] was used to search sequence homologs for the domain IDs of each peptidase family, obtained from the Pfam database [44]. Several protease families were identified (Table 1, Figure 1) in the wheat proteome.

In total, 1544 proteases were discovered: 459 cysteine proteases from 12 families, 275 metallo-proteases from 17 families, 336 aspartic proteases from two families, 446 serine proteases from five families, and 28 threonine proteases.

The С1 family consisted of 181 members. This family was represented by the C1A subfamily of papain-like cysteine proteases (PLCPs) according to MEROPS nomenclature [45]. It is noteworthy that some proteases were simultaneously annotated with two distinct Pfam IDs: Peptidase_C1 and Peptidase_C1_2 with E value in the order of e-100 and e-10, respectively. These peptidases form an independent group with no homologs characterized across *A. thaliana* papain-like cysteine proteases (PLCPs). Peptidase_C1_2 corresponds to the C1B subfamily (according to MEROPS nomenclature). Thus, it was concluded that the *T. aestivum* genome contained no C1B peptidases. Nevertheless, C1 peptidases formed two distinct branches on the phylogenetic tree, divided by several proteases from families C2 and C78, indicating possible divergent processes within the C1A subfamily of cysteine proteases (Figure 1, “Cysteine”). There were some peculiarities on this vast tree of distantly related cysteine proteases. For example, sequences annotated to the C48 family according to its Pfam domain were widely “dispersed”, whilst some “C48 sequences” were closer to other families than to each other (Figure 1, “Cysteine”).

Serine proteases formed distinct family groups containing close homologs, whereas the other groups were distant from each other (Figure 1, “Serine”). Metallo-proteases were represented in a variety of families (Table 1), which were mixed up on the phylogenetic tree (Figure 1, “Metallo-proteases”). Proteases from M1 and M3 families occurred in different parts of the tree, whereas M10, M16, M24, M28, M41, and M48 formed distinct. Aspartic proteases from different families (the A1 family and A22B subfamily) did not align properly with each other, indicating an independent evolutionary relationship within the group of aspartic proteases (Figure 1, “Aspartic”). Proteases from the A1 family are unique for *T. aestivum* and are composed of *Triticum aestivum* xylanase inhibitor N (TAXi_N) and TAXi_C domains, which are distant but homologous to each other’s domains. It is noteworthy that the TAXi_N domains from different proteases are closer to each other than the TAXi_N and TAXi_C domains within one protease. Thus, the phylogenetic tree (Figure 1, “Aspartic”) indicates the divergent processes between the TAXi domains.

The phylogenetic trees based on the alignment of relevant domains were constructed for families currently known to be associated with plant immunity such as C1, C13, C14 (Figure 2), and S8 (Figure 4) [34]. Domain structures were identified for all proteases. Some criteria were introduced to demarcate discrete subgroups the following criteria were used based on the “from leaves to roots” principle:Check the common node for two leaves. If the bootstrap value for this node is more than 70, then merge these leaves into one subgroup. Otherwise, these leaves should be considered as distinct, independent groups. Repeat this step as many times as needed.Check the common node for the node with a low bootstrap value. If the bootstrap value for the upper node is more than 70, then subgroups should be considered as members of a larger group. However, the topology of tree nodes with a low bootstrap value cannot be resolved, although a highly credible, common node clearly indicates the shared origin of these subgroups.If uniquely specific features are shown as representative of some subgroups, they should be considered as independent.

As proteases from the C1A, C13, and C14 families are closely related, a single phylogenetic tree was constructed for these families (Figure 2). In total, 25 clusters of C1A proteases were identified. Nine clusters corresponded to the nine groups identified by Richau et al. [46] for the A. thaliana PLCP subfamilies (Figure 2): Responsive to Dehydration 21-like (RD21), Xylem cysteine peptidase 2-like (XCP2), Cysteine EnsoPeptidase 1-like (CEP1), Senescence-Associated Gene 12-like (SAG12), Xylem Bark Cysteine Peptidase 3-like (XBCP3); and Responsive to Dehydration 19-like (RD19), Arabidopsis Aleurain-like Protease-like (AALP), cathepsin B-like and THIol protease 1-like (THI1). In addition, protease groups homologous to those from rice (*Oryza sativa*) were also identified (indicated as *O. sativa* cysteine proteases (OsCPs) in Figure 2). Moreover, proteases from other groups had previously undescribed homologs (shown in red in Figure 2). Several of the homologous proteases closest to some of these newly identified groups were indicated (Figure 2): 16 were COTyledon abundant protease 44-like (COT44), 19 were vignain-like and 22 were Senescence-Associated Gene 39-like (SAG39).

Typically, C1A proteases consist of a prodomain and catalytic domain, and, in the case of XBCP3-like and RD21-like proteases, a granulin domain (Figure 3). Some of the other clades possess distinct features: group 15 contains proteins with Domain of Unknown Function 4371 (DUF4371); and the SAG39-like cluster (group 22) contains the No Apical Meristem-associated (NAM-associated) domain.

The C13 peptidases indicated on the phylogenetic tree (group 26, Figure 2) were classified into five closely related groups encompassing the homologs of VPEs α, β, γ, and δ. Analysis of the domain architecture of proteases from the C13 family revealed only the presence of the catalytic domain. Metacaspases from the C14 family formed three clades, which included Type I and Type II metacaspases. Type I metacaspases had an N-terminal zinc-finger domain (zf-Lysine-specific histone demethylase 1 (LSD1)), which coincided with the already described features of Type I metacaspases [46].

Eighty-two serine proteases from the S8 family, also known as subtilases, were classified into nine groups, for which the closest homologs from *A. thaliana* were identified: 1—subtilase 3.8-like (SBT3.8-like), 2—SBT1.7-like, 3—SBT1.4-like, 4—SBT5.3-like, 5—CO_2_-response secreted protease-like, 6—SBT1.8-like, 7—SBT1.7-like, 8—SBT2.5-like, 9—tripeptidyl peptidase 2-like (Figure 4). Proteases from almost all the groups carried not only a catalytic domain (Peptidase_S8), but also a conservative prodomain (Inhibitor_I9, Figure 5). Groups three to six and eight contained proteases with a Protease-associated (PA) domain. Subtilases from groups eight and nine contained a fibronectin Type III-like (fn3_5) domain.

### 2.2. Differences in Degradomes of Two Wheat Cultivars Were Revealed

Using LC-MS data, proteases present in healthy plants of two cultivars, Khakasskaya and Darya, were quantified on the basis of full-specific peptides (full-tryptic and full-AspN peptides) [47]. This indicated that the degradome of wheat varies within the species (Figure 6). According to a comparative analysis, the Khakasskaya and Darya cultivars of wheat express an almost equal number of proteases (94 and 79 proteases, respectively) and share 49 proteases, representing 52,1% and 62% of the overall number of proteases in Khakasskaya (Appendix A) and Darya (Appendix A) cultivars, respectively. Serine proteases are relatively abundant in both cultivars: half that amount of cysteine, metallo- and aspartic proteases were detected. The most prevalent families of proteases in both cultivars are C1, S8, S9, S10, M20, M24, and A1 (Appendix A).

### 2.3. Infection by Pathogens Leads to an Increase in the Number of Expressed Proteases

Based on the LC-MS approach, a comparison was made of proteases expressed in healthy plants and those infected by *P. recondita* (biotrophic pathogen) and *S. nodorum* (necrotrophic pathogen) plants at 24 h post inoculation (hpi). Identification of proteases was based on quantification of full-specific tryptic and full-specific AspN peptides. An increased number of detected proteases was found upon both infections (Figure 7): 117 and 77 proteases were detected in plants infected by *P. recondita* (Appendix A) and *S. nodorum* (Appendix A), respectively, which share 55 (58,5%) and 45 (60%) proteases with controls (healthy Khakasskaya (Appendix A) and Darya (Appendix A) plants, respectively).

In the case of *P. recondita* infection, induction of proteases occurred mostly from families C1, C13, C48, C65, M3, M41, M20, M24, M17, S8, S9, S10, A1, whilst in the case of *S. nodorum* infection, induction occurred mostly from families C1, C14, C26, C48, C65, M3, M41, M24, S8, S9, S10, A1 (Appendix A).

### 2.4. The Pool of Substrates Cleaved by Proteases In Vivo Expands upon Both Types of Infection

On the basis of these LC-MS data, a search was undertaken for potential substrates cleaved by proteases in healthy plants of both cultivars. This was done through identification of semi-specific peptides released after digestion with trypsin, or AspN proteases used in the present study. One terminal of the peptide corresponds to a specific hydrolysis site (cleavage after R and K for trypsin, and before C and D for AspN), and the other corresponds to non-specific hydrolysis sites. It is worth stating that semi-specific peptides may arise from two main sources: as a result of hydrolysis by endogenous proteases within the sample, or as a result of the truncation of regular tryptic peptides through in-source fragmentation (ISF) within the electrospray ionization (ESI) source [48]. The impact of ISF in generation of semi-specific peptides was assessed earlier, and it depended to a large extent on the complexity of the biological sample, ranging from 1% (in a mouse-brain sample) to 57% (in a standard protein mixture) of the total amount of tryptic peptides (including both full-specific and semi-specific peptides) [48]. How ISF affects truncation of the peptides in plant samples has not yet been analyzed. However, despite the described limitations, identification of semi-specific peptides was originally used to analyze complex protein samples in shotgun proteomics [49]. Identifying more peptides, e.g., non-tryptic peptides, may increase the peptide coverage and improve protein identification and/or quantification. Moreover, the semi-specific peptide approach identified processing patterns of N-terminal signal peptide in human aspartyl-tRNA synthetase, by combining immuno-enrichment of the protein samples with classical shotgun LC-MS/MS analysis and semi-tryptic database searching [50]. Similarly, as semi-specific peptides are likely to be a product of in vivo hydrolysis by endogenous proteases, this method may be successfully used for studying both the maturation of proteases in vivo and their involvement in cellular proteolytic cascades.

The search for potential substrates of endogenous proteases in healthy plants revealed an almost equal number of cleaved substrates: 220 (Appendix A) and 178 (Appendix A) proteins were identified (85 shared) in healthy Khakasskaya and Darya plants (Figure 8A). The presence of substrates presumed to be cleaved by caspase-like and metacaspase-like proteases was also investigated. As stated earlier, plant immune responses and PCD are associated with caspase-like and metacaspase activities, which include hydrolysis of substrates at very specific sites such as VEID, DEVD and YVAD sites. However, it should be noted that there are limited number of known proteolytic sites for such proteases and overall substrate specificities of caspase-like proteases [51]. Most metacaspases also remain uncharacterized. A generalized rule for identifying substrates was subsequently applied: XXXD for caspase-like and XXXR, XXXK for metacaspase-like proteases, where X is any amino acid. The number of probable substrates of caspase-like proteases was found to be almost equal amongst healthy Khakasskaya and Darya plants (10 and nine, respectively, with five shared), whereas slightly more substrates of metacaspase-like proteases were found in healthy Khakasskaya and Darya plants (11 and 19, respectively, with eight shared; Figure 8A).

As the number of expressed proteases increased upon both infections (see previous section), the number of substrates cleaved by endogenous proteases was checked to determine whether they also increased upon infection. A significant increase in the number of substrates was observed. In the case of *P. recondita* infection (Figure 8B, Appendix A) this number increased from 220 to 518 (121 shared), in the case of *S. nodorum* infection (Figure 8C, Appendix A)—from 178 to 300 (77 shared. Substrates of caspase-like and metacaspase-like proteases were also detected. Both healthy and infected plants are characterized by increased number of proteins cleaved at XXXD sites. 10 and 28 (four shared) sites were identified in the case of *P. recondita* infection (Figure 8B); nine and 15 (three shared)—in the case of *S. nodorum* infection (Figure 8C). In addition, both types of infection are characterized by an increase in metacaspase-like activity. 11 and 22 (seven shared) sites were identified in the case of *P. recondita* infection (Figure 8B); 19 and 17 (11 shared) sites—in the case of *S. nodorum* infection (Figure 8C).

Potential substrates of caspase-like and metacaspase-like proteases cleaved in response to infections were also investigated to ascertain which were caused by both *P. recondita* and *S. nodorum* 24 hpi. Among these substrates, seven caspase-like and three metacaspase-like substrates were found cleaved upon both infections. These substrates are summarized in Table 2.

### 2.5. Infection Induces Synthesis and/or Proteolytic Activation of Proteases through Recruitment of Proteases Other Than Caspase-Like and Metacaspase-Like Proteases

It has been shown that caspase-like and metacaspase-like activities are present in wheat cells after infection by both *P. recondita* and *S. nodorum*. The impact of such activities on proteolytic activation of all proteases was examined in healthy plants and those infected by both pathogens 24 hpi. To do this, the same approach was implemented, based on identification of semi-specific peptides after digestion by trypsin or AspN for LC-MS. A search was undertaken for peptides mapped around the bordering prodomain-proteolytic domain (±40 amino acids from the border), since a large number of proteases contain autoinhibitory prodomain, which needs to be cleaved to obtain an active protease. In the case of metacaspases, semi-specific peptides at known processing sites for homologous enzymes [46] were selected. The processing status of the proteases was, therefore, identified in healthy plants and those infected by *P. recondita* 24 hpi (Table 3, Appendix A) and *S. nodorum* (Table 3, Appendix A).

Upon biotrophic infection, a significant decrease in the number of processed proteases was detected, in comparison to the control (Table 3). Healthy and infected plants share six processed proteases (Appendix A); eight proteases from families M17, M20, M24, M41, S8, S10, A1 were detected, presumably activated upon infection (Appendix A); healthy plants are characterized only by 1 unique protease (Appendix A). Upon necrotrophic infection, three processed proteases were detected both in healthy and infected plants (Appendix A). However, infection caused processing of more proteases from families C1, C13, S8, S49, M24, A1 (six proteases; Appendix A) in comparison to the control (four proteases; Appendix A).

It is worth mentioning that no recognition sites of metacaspases (XXXR, XXXK) was found among detected activation sites of proteases (Table 3). Moreover, the number of caspase-like sites (XXXD) has not changed upon *S. nodorum* infection (1 activation site, shown in Table 3, Appendix A) and slightly increased after infection with *P. recondita* (from zero to three, shown in Table 3, Appendix A).

## 3. Discussion

In total, 1,544 proteases were found, encoded in the whole genome of *T. aestivum*. This constitutes a significant number of encoded proteases, for example, when compared with diploid dicotyledon plants, such as *Nicotiana benthamiana* L., with predicted proteases measured at around 1,245. Similarly, 796 proteases have been found in *A. thaliana* and 901 in tomatoes. Monocotyledon plants, such as rice, contain 997 [20].

The wheat degradome is represented by a large number of protease families, which have been subclassified into various groups. The most striking example is the C1A subfamily, i.e., PLCPs. In this group, 181 members were identified. With *A. thaliana*, 31 proteases were reported [52], 33 in rice [17], 43 in rubber [53] and 33 in papaya [43]. Subfamilies of PLCPs (already described for *A. thaliana* [52] and rice [17]) were also identified. Moreover, given that no known homologs from other organisms were found, eight groups of PLCPs were defined as being unique to the species subfamilies (Figure 3, in red). Classification of 82 subtilases (S8 protease family) resulted in identification of groups homologous to *A. thaliana* proteases and their features have been reported earlier [54,55,56,57,58].

LC-MS has shown that different sets of proteases can be detected in different wheat cultivars. Proteomes of Khakasskaya and Darya cultivars share 58,5% and 60% of all proteases, respectively, which is slightly less than already reported difference of 79.3% between proteomes of wheat cultivars [2,3].

The use of LC-MS in this study has shown for the first time that biotrophic and necrotrophic infections induce expression of proteases in plants and that these proteases are different. Responses included induction of M17 and M20 proteases in the case of biotrophic infection, and C14, C26 proteases in necrotrophic infection. It should be noted that most proteases induced by both biotrophic and necrotrophic pathogens are attributed to the same families, such as C1, C13, C48, C65, M24, M41, S10, S9, S8 and A1. The same C13 protease was activated upon both infections, but at different sites (TRIAE_AA0544900.2). C13 proteases include VPEs previously associated with responses to viruses [27] and *P. syringae* [25], which are biotrophic pathogens as well as *P. recondita*. C14 proteases include metacaspases, already widely proven to mediate plant immune responses [46]. The M17, M20, M24, M41, C26 protease families remain to be uncharacterized groups of proteases. It has been concluded that these families are likely to be associated with a response to different types of biotic stress in wheat.

A novel approach has been proposed in this study for the identification of in vivo proteolysis events, based on a search for semi-specific peptides in LC-MS data. This has facilitated assessment of overall proteolytic activity for the first time in healthy and stressed plants. These results indicate that infections by *P. recondita* and *S. nodorum* pathogens are associated with an increase of between one and a half to two times the number of substrates potentially cleaved by endogenous proteases in plant cells. However, the assessments undertaken in this study were qualitative (not quantitative), precluding assumptions about the physical amount of each substrate that was possibly cleaved. Nevertheless, it can be concluded that the number of proteins cleaved increased significantly in response to biotic stress in comparison to healthy plants, indicating the involvement of proteolytic cascades.

Identification of substrates for caspase-like and metacaspase-like proteases revealed that, despite a significant increase in the overall number of proteins cleaved after induction of biotic stress, the quantity of substrates for caspase-like proteases increased slightly. However, substrates for metacaspase-like proteases increased from one and a half to two times in infections of both pathogens. This result confirms the involvement of metacaspases in plant response to pathogens described earlier [29,32,58] and either indicates that caspase-like proteases were not active at 24 hpi and PCD-associated processes were not induced, or that caspase-like enzymes are not implemented in response to biotic stress in wheat.

It is interesting that upon infection by different pathogens in different wheat cultivars, the same substrates were cleaved by caspase-like and metacaspase-like proteases (Table 2) such as receptor-like protein kinase, permease, disease resistance proteins, and other proteins. Identified receptor-like protein kinases may be directly involved in the establishment of microbe-associated molecular patterns (MAMP)-triggered immunity (MTI) or effector-triggered immunity (ETI) [59]. The relationship between such a receptor-like protein kinase and the protease that cleaves it, is a completely unstudied field of research in plant immunity. One of the few known examples is that of ectodomain shedding, which has been identified as cleavage of the chitin receptor of *A. thaliana*, Chitin Elicitor Receptor Kinase 1 (CERK1), by an obscure protease [60]. These substrates may become potential targets for further research. The other substrate—FAR1-like protein—was found to be cleaved by both caspase-like and metacaspase-like proteases. In *Arabidopsis*, it was associated with light control and plant development [61]. The role of this protein in plant immunity needs to be studied.

The pool of substrates cleaved by metacaspase 9 from *A. thaliana* has been identified earlier [62]. It can be seen that some substrates are attributed to the same class of enzymes, for example, helicases. However, it is worth mentioning that cellular pathways are more conservative than the proteins involved in them. Hence, differences in identified potential substrates of metacaspases may be due to the fact that wheat is a monocot organism, whereas *A. thaliana* is a dicot. This study has found some potential substrates that are completely uncharacterized proteins. This result, therefore, requires confirmation and more targeted research.

The use of LC-MS analysis has made it possible to define proteolytic activation of proteases in vivo. These findings show that proteases from M17, M20, M41, S10, A1 families are likely to have been activated upon biotrophic infection, as opposed to the S49 family in the case of necrotrophic infection. The core proteases activated by both infections are from C1, C13, S8, M24, and A1 families. The C1 family includes PLCPs, C13 is a VPEs containing family, S8 family includes subtilases associated with plant immunity, which have been well-characterized in previous research [34]. However, M24 and A1 are very poorly described families of proteases, linked to plant immune responses for the first time in this study.

Our results indicate that in infected wheat, most proteolytic sites located within proteases significantly differ from the sites commonly recognized by caspases or metacaspases. Only 0/7 and 1/7 caspase-like cleavage sites (XXXD) were detected within the proteases of healthy Khakasskaya and Darya plants, respectively, whereas in infected plants, 3/16 and 1/9 cleavage sites were found for *P. recondita* and *S. nodorum* infections, respectively. No activation sites recognized by metacaspases were identified in neither healthy, nor infected by pathogens samples.

Data in this study suggest that a number of proteases are presumably activated upon both types of infection. However, this activation requires conditions other than those of caspase-like or metacaspase-like activities. This is a striking result, which gives an insight into the previously unexplored field of proteolytic cascades in plants and wheat, in particular. This study attempted to provide a comprehensive investigation of proteolytic cascades, triggered in by *P. recondita* and *S. nodorum* infections 24 hpi in wheat, but these proteases remain uncharacterized and require further consideration.

## 4. Materials and Methods

### 4.1. Phylogenetic Analysis of the Wheat Protease Families

A complete set of protein sequences for *T. aestivum* (*n* = 154,140) was obtained from the Ensembl genome database (ftp://ftp.ensemblgenomes.org/) [63]. The most recent release (release 39) was used. Protein sequences shorter than 200 amino acid residues were omitted. Proteins that contained more than four unspecified amino acid residues (i.e., XXXX, where X is any amino acid residue) were also omitted. Any sequences in the data set differing by less than 5% of the nucleotide sequence were omitted. This curated data set (*n* = 66,615) was analyzed using the HMMER 3.1b2 package (http://eddylab.org/software/hmmer/hmmer-3.1b2.tar.gz) [64] in order to obtain the distribution of Pfam domains in the wheat proteome.

Protein sequences were divided into protease families according to their Pfam domain annotation: “Peptidase_C” was considered as a cysteine peptidase; “Peptidase_M” was considered as a metallo-protease; “Peptidase_S” was considered as a serine peptidase; “Peptidase_A” and “TAXi_” were considered as aspartic peptidases; and “proteasome” was considered as a threonine peptidase. The sequences containing these domains were analyzed by the HMMER 3.1b2 package using the E-value threshold of 10^−4^. Relevant domains were excised from complete protein sequences. Various threshold lengths were applied for different protease families: 200 amino acid residues for cysteine and serine protease families; 150 amino acid residues for metallo-protease aspartate protease families; and 100 amino acid residues for threonine protease families. Relevant domain sequences were aligned using the MAFFT server (https://mafft.cbrc.jp/alignment/software/) [65]. The evolutionary history was inferred using the neighbor-joining method implemented in MEGA7 (https://www.megasoftware.net/) [66]. The confidence in the trees [66] was estimated by 100 bootstrap replicates. Bootstrap values greater than 70 are shown in Figure 2 and Figure 3. Phylogenetic trees were visualized with FigTree (v. 1.4.2) (http://tree.bio.ed.ac.uk/software/figtree/).

### 4.2. Plant Growth and Infection by Different Pathogens

Wheat seeds from the universally susceptible Khakasskaya and Darya lines were washed with water and held for 15 minutes (min) in a 5% solution of KMnO_4_, then placed on wet filter paper in Petri dishes and incubated for two days at 22 °C. Sprouted seeds were planted in vegetation vessels with earth. The plants were grown in a climatic chamber for eight to 10 days with a 16-h light schedule, at a temperature between 20 °C and 22 °C. Plants with a fully-developed, first real leaf were used for the experiments.

A culture of the *Stagonospora nodorum* fungus (Berk; strain B-24/MS2 from the State Collection of Phytopathogenic Microorganisms) was grown on a sterile wheat grain [67]. A shaking flask was filled to two thirds of its volume with grain and water was added, corresponding to half the weight of the grain. The flask was autoclaved for one hour (0.5 atm, 114 °C). The resulting mass of grain was cooled and infected with pieces of mycelium collected from the surface, then covered with mycelium from the fungus of the agar medium in Petri dishes. The flasks of grain were incubated with a thermostat set at 26 °C for seven days. Spores were washed from the surface of the grain with distilled water. Spore suspension in a concentration of 1 × 10^6^ spores/mL was used in the experiments. Spore suspension of *S. nodorum* fungus was used to infect wheat plants of the Darya line. The suspension was applied by means of an atomizer: consumption—10 mL/100 plants. Tween-40 was added dropwise to the suspension in 100 mL of slurry. The inoculated plants were kept in a moist chamber for two days at a temperature between 20 °C and 22 °C, then returned to the climatic chamber with the original regime.

Samples of brown rust (*P. recondita* Rob. ex Desm f. sp. Tritici; Lower Volga population 757) were suspended in water to a concentration of 1 × 10^6^ spores/mL. Tween-40 (one drop per 100 mL) was added. Wheat plants (Khakasskaya line) at the stage when the first real leaf had unfolded, were inoculated by rubbing a suspension of spore leaves until the surface was completely moistened and inoculated. The plants were then transferred to a moist chamber at a temperature between 20 °C and 22 °C. After two days, the plants were returned to the climatic chamber with the original regime [68].

### 4.3. Plant Sample Preparation for Mass Spectrometry

The chloroform/methanol precipitation method was used for extraction of total protein. Enzymatic hydrolysis of the proteins was performed according to the procedure previously described [69]. Three biological and three technical replicates were produced in each experiment.

### 4.4. LC-MS/MS Analysis

One microgram of peptides in a volume of 1–4 µL was loaded onto the Acclaim µ-Precolumn (0.5 mm × 3 mm, 5 µm particle size, Thermo Scientific, Rockwell, IL, USA) at a flow rate of 10 µL/min for 4 min in an isocratic mode of Mobile Phase C (2% acetonitrile, 0.1% formic acid). Then the peptides were separated with high-performance liquid chromatography (HPLC, Ultimate 3000 Nano LC System, Thermo Scientific, Rockwell, IL, USA) in a 15-cm long C18 column (Acclaim^®^ PepMap™ RSLC inner diameter of 75 μm, Thermo Fisher Scientific, Rockwell, IL, USA). The peptides were eluted with a gradient of buffer B (80% acetonitrile, 0.1% formic acid) at a flow rate of 0.3 μL/min. Total run time including initial 4 min of column equilibration to buffer A (0.1% formic acid), then gradient from 5–35% buffer B over 65 min, 6 min to reach 99% buffer B, flushing 10 min with 99% buffer B, and 5 min re-equilibration to buffer A amounted to 90 min.

Mass spectrometric analysis was performed at least in triplicate with a Q Exactive High-Field (HF) mass spectrometer (Q Exactive HF Hybrid Quadrupole-OrbitrapTM Mass spectrometer, Thermo Fisher Scientific, Rockwell, IL, USA). The temperature of capillary was 240 °C and the voltage at the emitter was 2.1 kV. Mass spectra were acquired at a resolution of 120,000 (MS) in a range of 300−1500 mass-to-charge ratio (*m*/*z*). Tandem mass spectra of fragment were acquired at a resolution of 15,000 (MS/MS) in the range from 100 *m*/*z* to *m*/*z* value determined by a charge state of the precursor, but no more than 2000 *m*/*z*. The maximum integration time was 50 ms and 110 ms for precursor and fragment ions, respectively. Automatic Gain Control (AGC) target for precursor and fragment ions were set to 1 × 10^6^ and 2 × 10^5^, respectively. An isolation intensity threshold of 50,000 counts was determined for precursor’s selection, and up to top 20 precursors were chosen for fragmentation with high-energy collisional dissociation (HCD) at 29 Normalized Collision Energy (NCE). Precursors with a charged state of +1 and more than +5 were rejected and all measured precursors were dynamically excluded from triggering of a subsequent MS/MS for 20 s.

### 4.5. Protein Identification and Determination of Sites Hydrolyzed In Vivo by Endogenous Proteases

For the analysis of the mass spectrometry data, a database was built using *T. aestivum* proteome (release 39, Ensembl genome database (ftp://ftp.ensemblgenomes.org/) [63]) merged with pathogen proteomes. In the case of *P. recondita* infection, assembly *Puccinia triticina* 1-1 BBBD Race 1 (NCBI ID 1628) [70] was used, while in the case of *S. nodorum* infection, *Parastagonospora nodorum* SN15 (assembly ASM14691v2) [71] was used. In our study, we performed 3 biological and 3 technical replicates for each experiment, however, we used only 2-3 biological 2-3 technical replicates each. It depended on the fact that some of the replicates were not successful and, thus, were omitted.

Raw data were processed using IdentiPy search algorithm [47]. Peptide scoring was based on Hyperscore from X!Tandem (Appendix A) [72]. IdentiPy algorithm suggests autotune feature that allow to reprocess spectra and to optimize initial search parameters. Initial values for parameters were set by default as 100 ppm for precursor mass error, 500 ppm for fragment mass error, and 5 for allowed miscleavages. The results of preliminary searching were filtered to 1% False Discovery Rate (FDR) using the target-decoy approach and analyzed statistically to derive the optimal parameters that were adjusted by the algorithm for all searches performed in this study, 20 parts per million (ppm) for precursor mass error, 10–11 ppm for fragment mass error, and 2 for allowed miscleavages.

IdentiPy was used for protein identification. Post-search analysis and FDR filtering relied on multiparameter (MP) score algorithm [73]. Peptide-Spectrum Matches (PSMs), peptides, and proteins were validated at a 1.0% FDR estimated using the decoy hit distribution. Only proteins having at least two unique peptides were considered as positively identified.

In order to reveal differences in number of proteases in cultivars and between healthy and infected plants, full-specific (full-tryptic and full-AspN) peptides were identified in sequences from the database containing all 1544 wheat proteases (Appendix A). Proteases that were covered by at least two unique peptides that were present in all biological and all technical replicates for each type of samples were taken for further analysis.

In order to identify the sites of in vivo hydrolysis in all proteins from the samples, the search of semi-specific (semi-tryptic and semi-AspN) peptides in the sequences from the database containing whole wheat proteome was conducted (Appendix A). Semi-specific peptides present in all biological and all technical replicates for each type of sample were taken for further analysis.

In order to identify the sites of in vivo processing of proteases, the search of semi-specific peptides within the areas that contained 40 amino acid residues, extracted before and after the first peptidase domain amino acid site, was conducted (Appendix A). For this search, the database containing wheat proteases that were identified with the use of full-specific peptides (Appendix A) were used. In the case of metacaspases, known sites [33] were used for the search. Peptidases containing semi-specific peptides (100% identical, 100% covered), released after digestion by AspN or Trypsin, as defined by IdentiPy, were identified using the BlastP algorithm [74].

## 5. Conclusions

This article has characterized the degradomes of wheat. Comprehensive analysis of the *T. aestivum* genome revealed a relatively large number of proteases (1544 in total) belonging to the five major protease families: serine, cysteine, threonine, aspartic, and metallo-proteases. Unique protease groups from the C1 family were identified. Analysis of the LC-MS data obtained for degradomes of distinct wheat cultivars (Khakasskaya and Darya) revealed significant differences ~40%, equally distributed between different catalytic types of proteases, which could indicate the complementation of different proteases by one another. These findings underline the importance of interactions between components of the degradomes of any organism and wheat, in particular. In turn, these should provide insights into the physiology and biochemistry of plant immunity about which little is currently known. Infection by biotrophic (*P. recondita*) or necrotrophic (*S. nodorum*) pathogens induced drastic changes in the presence of proteolytic enzymes, as observed in the LC-MS data. However, the immune response is associated with the same core, consisting of proteases from C1, C13, C65, M16, M50, S8, S10, and A1 families. This indicates that the immune response is associated with well-known proteases. However, numerous uncharacterized protease families still require careful consideration and study. Infection by both pathogens enhances overall proteolytic activity in wheat cells. The potential for proteolysis in vivo by endogenous protease substrates increases in response to both types of infection. Biotic stress seems to activate proteolytic cascades and overall degradation of proteins. Analysis of protease-derived peptides detected by LC-MS revealed proteolysis sites, which are probably the targets of autocatalytic activation or hydrolysis by other proteases within the proteolytic cascades, involving neither caspase-like proteases nor metacaspases. However, this needs to be corroborated using activity-based approaches to validate the method for determining processed proteases.

## Figures and Tables

**Figure 1 ijms-19-03991-f001:**
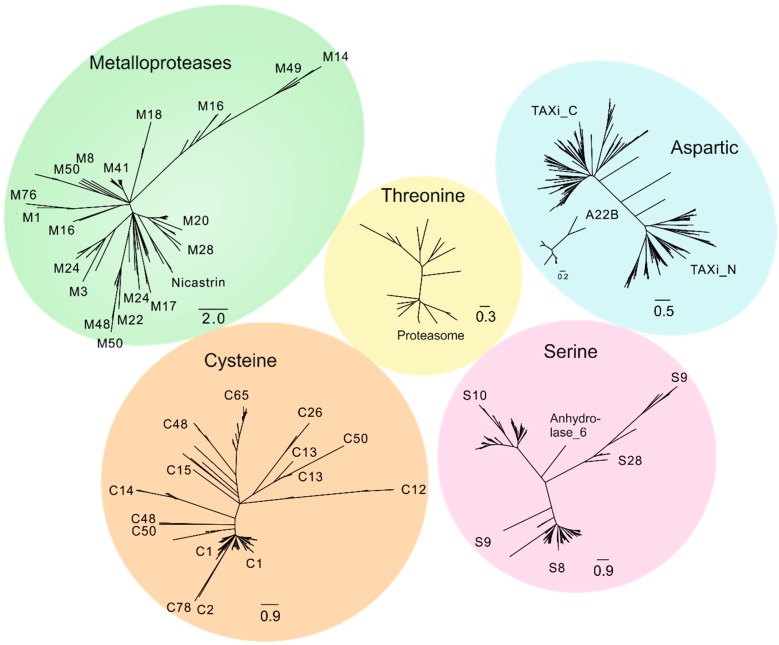
The *T. aestivum* degradome landscape. Metallo-proteases are indicated in green, threonine–in yellow, aspartic–in blue, cysteine–in orange, serine–in purple. Unrooted phylogenetic trees display diversity across protease families. The trees are drawn to scale, with branch lengths in the same units as those of the evolutionary distances used to infer the phylogenetic tree. Evolutionary distances were computed using the p-distance method [45] and their units correspond to the number of amino acid differences per site.

**Figure 2 ijms-19-03991-f002:**
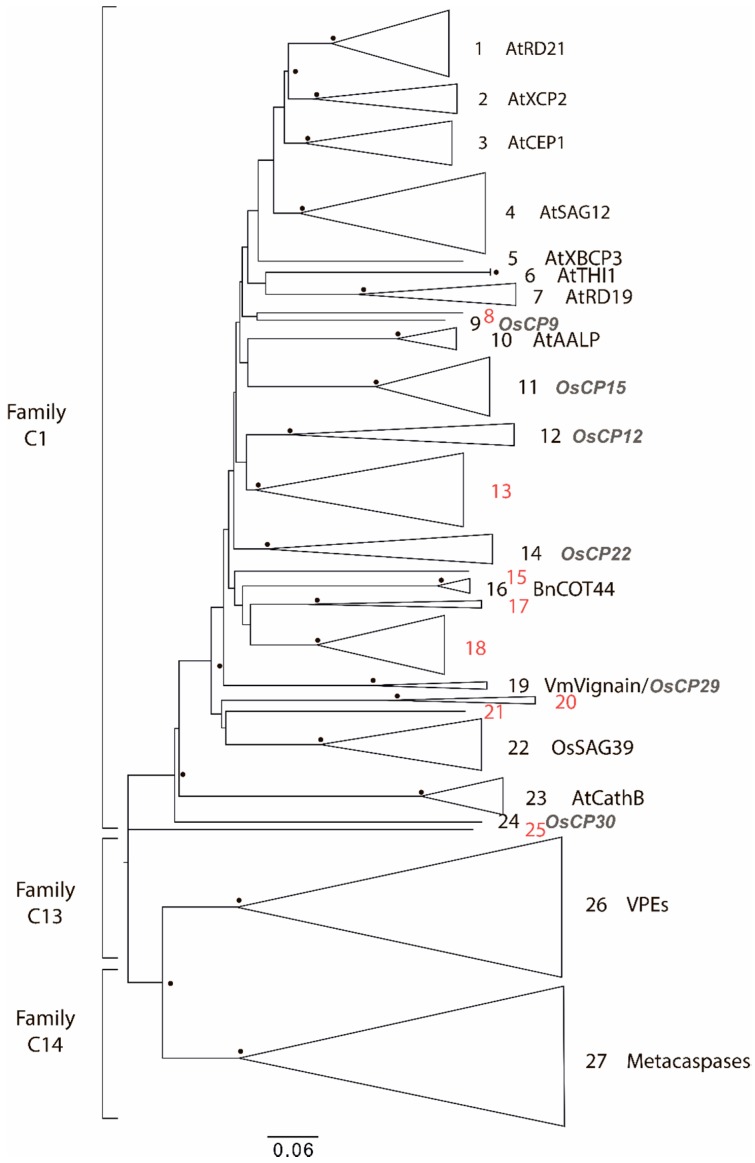
Phylogenetic tree of families C1, C13, and C14 of cysteine proteases. Triangles represent the contracted clades and the size of each protease group. Bootstrap values above 70% are shown with black circles at the relevant nodes. The closest homologs of the groups of proteases are indicated on the right of the phylogenetic tree: At—*A. thaliana*, Os—*Oryza sativa*, Bn—*Brassica napus* and Vm—*Vigna mungo*. Unique for *T. aestivum* groups are shown in red.

**Figure 3 ijms-19-03991-f003:**
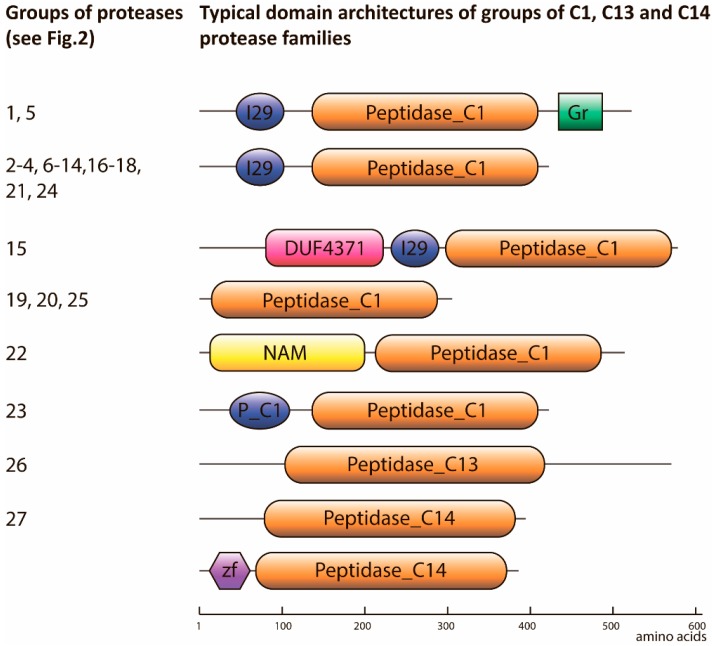
Typical domain organization of proteases from families C1, C13, and C14. The numbers of groups correspond to those indicated in Figure 2. The length of all the schemes are indicated on a scale according to the bar below the schemes. Pfam ID: Peptidase_C in orange—proteolytic domain of families C1, C13, and C14 of proteases; I29 in blue—Inhibitor_I29; NAM in yellow—NAM-associated; P_C1 in blue—Propeptide_C1; zf in purple—zf-LSD1 domains.

**Figure 4 ijms-19-03991-f004:**
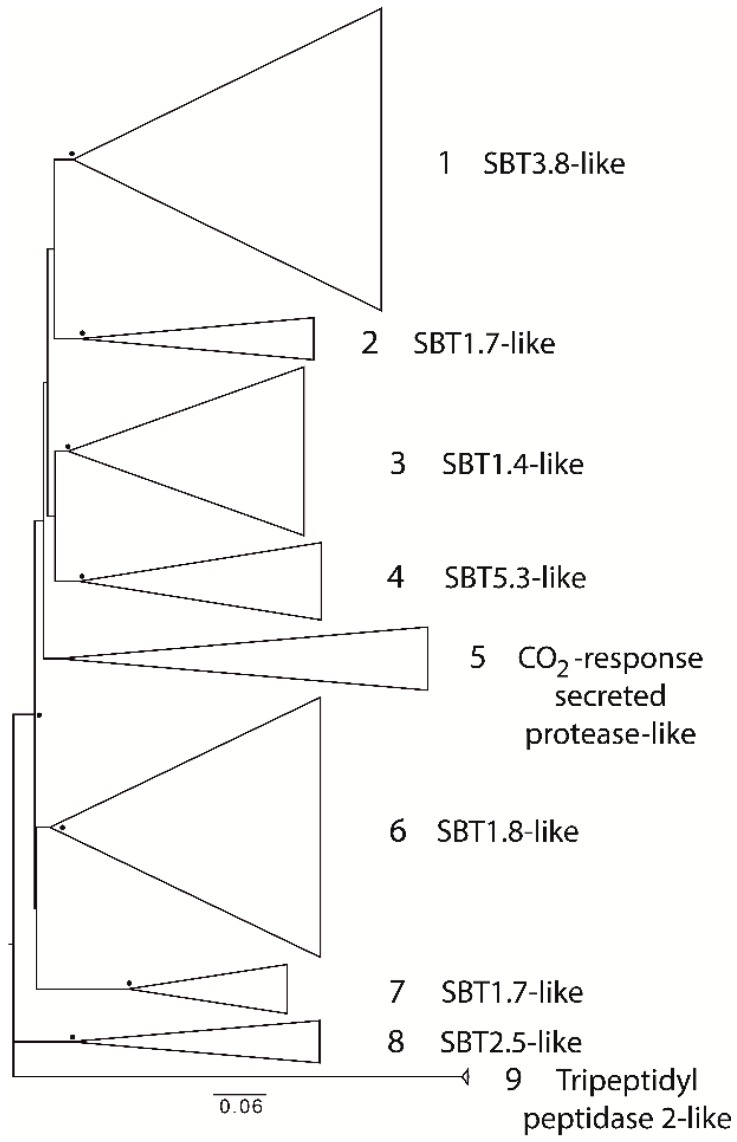
Phylogenetic tree of S8 family of proteases from wheat. Triangles represent the contracted clades and the size of each protease group. Bootstrap values above 70% are shown with black circles at the relevant nodes. The closest homologs of the groups of proteases are indicated on the right.

**Figure 5 ijms-19-03991-f005:**
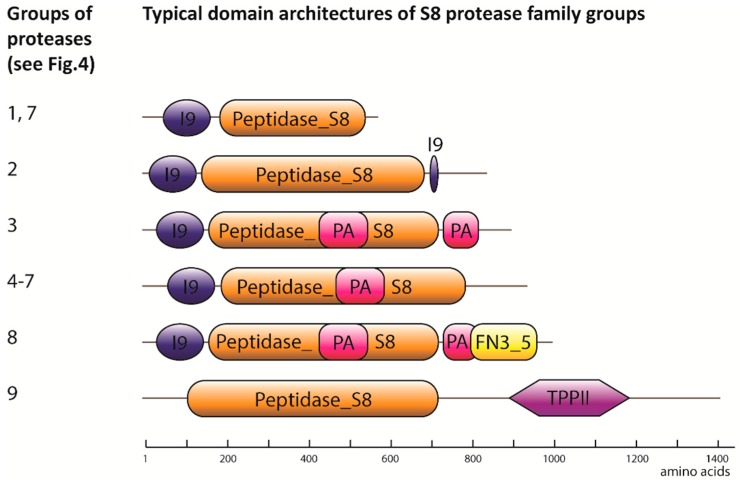
Domain architecture of S8 proteases from wheat. The numbers of groups correspond to those indicated in Figure 4. The length of all the schemes are indicated on a scale according to the bar below the schemes. Pfam IDs: Peptidase_S8 in orange—proteolytic domain of S8 proteases; I9 in blue—Inhibitor_I9; PA in magenta—Protease-associated; FN3_5 in yellow—fibronectin Type III-like; TPPII in purple—Tripeptidyl Peptidase 2 domains.

**Figure 6 ijms-19-03991-f006:**
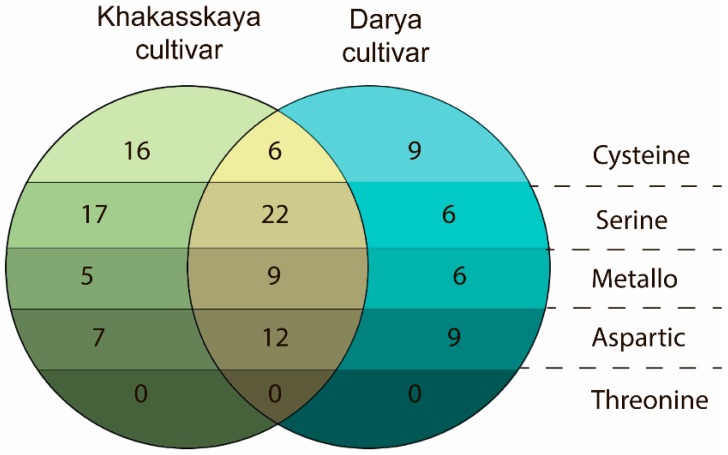
Venn diagrams representing the number of reproducibly quantified proteases for Khakasskaya (green shades) and Darya (cyan shades) wheat cultivars. Proteases found in both cultivars are indicated in yellow. Dotted lines divide proteases of different catalytic types.

**Figure 7 ijms-19-03991-f007:**
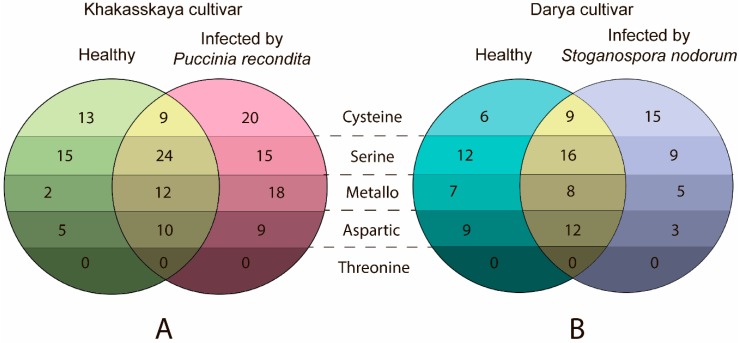
Venn diagrams representing the number of reproducibly quantified proteases in proteomes of (**A**) healthy Khakasskaya (green shades) plants and Khakasskaya plants infected by *P. recondita* (rose shades); (**B**) healthy Darya (cyan shades) and Darya plants infected by *S. nodorum* (purple shades). Proteases found in intersection between different samples are indicated in yellow.

**Figure 8 ijms-19-03991-f008:**
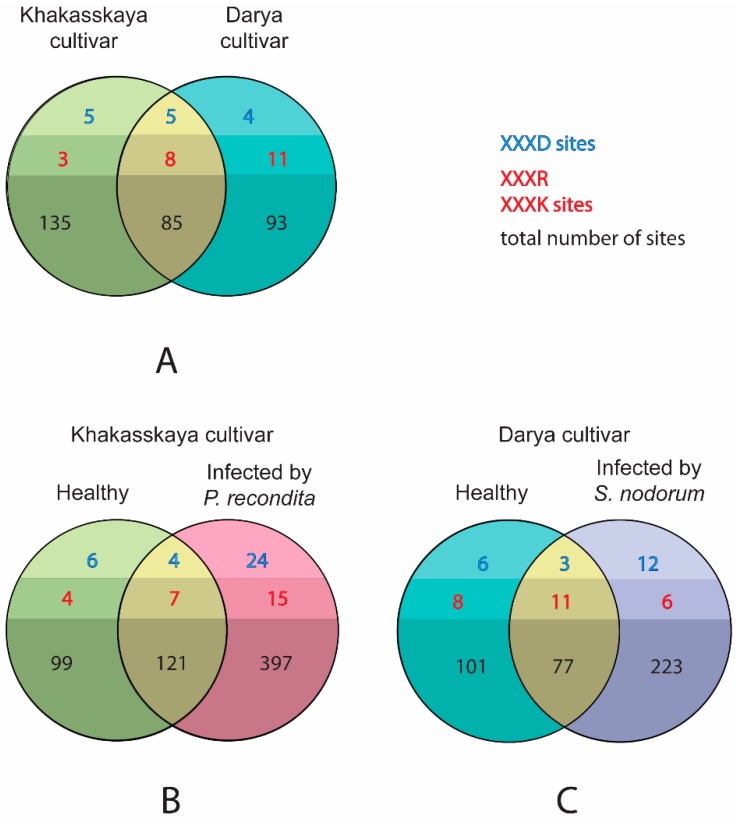
Venn diagrams representing the number of reproducibly quantified in vivo cleavage sites (detected with LC-MS) in all proteins across proteomes of (**A**) healthy Khakasskaya (green shades) and healthy Darya (cyan shades); (**B**) healthy Khakasskaya (green shades) and wheat plants infected with *P. recondita* (rose shades); and (**C**) healthy Darya (cyan shades) and wheat plants infected with *S. nodorum* (purple shades). Proteases found in intersection between different samples are indicated in yellow. XXXD—caspase-like site; XXXR, XXXK—metacaspase-like sites; X—any amino acid; numbers in bold and black—total number of cleavage sites across the proteomes; number of caspase-like sites are indicated by blue text, number of metacaspase sites—by red text.

**Table 1 ijms-19-03991-t001:** Representation of different protease families in the wheat genome.

Protease Family	Protease Families and Subfamilies (According to Pfam Nomenclature)
Cysteine	C1, C2, C12, C13, C14, C15, C26, C48, C50, C54, C65, C78
Serine	S8, S9, S10, S15, S28
Aspartic	A1, A22B
Metallo-proteases	M1, M3, M8, M10, M14, M16_M, M16_C, M17, M18, M20, M22, M24, M28, M41, M48, M49, M50B, M50
Threonine	T1

**Table 2 ijms-19-03991-t002:** Probable substrates of caspase-like and metacaspase-like proteases detected only in samples upon both *P. recondita* and *S. nodorum* infections. Annotation of proteins were made with Blast: the best hits with E-values < 1 × 10^−50^ are shown. The sites of hydrolysis are indicated by arrows.

Protein	Site of Hydrolysis
**Potential substrates of caspase-like proteases**	
Uncharacterized protein	VPTD↓AQLE
Protein Translocon at the inner envelope membrane of chloroplasts 22 (TIC 22), chloroplastic-like	ITLD↓QVYM
Disease resistance protein Rho-type GTPase-activating protein 2-like (RGA2)	VSAD↓GVTR
5′-3′ exoribonuclease 2-like	ILRD↓MVPL
Homeobox-DDT (DNA-binding homeobox-containing proteins and the different transcription and chromatin remodeling factors) domain protein Ringlet 3-like (RLT3)	KPED↓LTEY
Protein Fatty Acyl-CoA Reductase 1 (FAR1)-Related Sequence 5-like	LAAD↓HPRR
Constitutive Photomorphogenic1 (COP1)-interacting protein 7	IDID↓AELG
**Potential substrates of metacaspase-like proteases**	
Adenine/guanine permease Azaguanine Resistant 2 (AZG2)	CLAR↓TKSD
Wall-associated receptor kinase 5-like	LSTR↓NELI
Protein FAR1-Related Sequence 5-like	LFKK↓GVGA

**Table 3 ijms-19-03991-t003:** Proposed activation sites in proteases from different families based on LC-MS data in healthy Khakasskaya plants (exact sites and names of proteases are summarized in Appendix A); Khakasskaya plants infected with *P. recondita* (Appendix A); healthy Darya plants (Appendix A); and Darya plants infected with *S. nodorum* (Appendix A).

Catalytic Type of proteases	Family	Number of Detected by LC-MS Sites of Cleavage * (Number of XXXD Sites, If Any), [Number of XXXR or XXXK Sites, If Any]			
		Healthy Khakasskaya	Khakasskaya infected with *P. recondita*	Healthy Darya	Darya infected with *S. nodorum*
Cysteine	Peptidase_C1	2	1	1	2
	Peptidase_C13		1 (1)		1 (1)
Serine	Peptidase_S8		2	1	1
	Peptidase_S10		1		
	Peptidase_S49				1
Metallo-	Peptidase_M17		1		
	Peptidase_M20		1	1	
	Peptidase_M24		1		1
	Peptidase_M41	1	2	2	2
Aspartic	A1 (TAXi_C)	4	6 (2)	2 (1)	1
Total		7	16 (3)	7 (1)	9 (1)

* Each site of cleavage was found only in one protease, i.e., the number of detected cleavage sites coincides with the number of proteases in which they were found.

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
