# Peer review of "Proteomics Analysis Reveals That Caspase-Like and Metacaspase-Like Activities Are Dispensable for Activation of Proteases Involved in Early Response to Biotic Stress in *Triticum aestivum* L."

_ijms, 2018, doi:10.3390/ijms19123991_

Reviewer 1 Report

The manuscript of A.V. Balakireva et al. is dedicated to the analysis of host-pathogen interaction during a pathogen invasion. Proteomics analysis of protein’s degradome is an emerging topic allowing either to classify different pathogens by their peptide profiles, or to go further into details of a host defense mechanism revealing pathways activated under attack. Authors of this manuscript provide data from proteomics LC-MS/MS experiments, in which they give a detailed analysis of semi-tryptic peptides, which occur as a result of a molecular defense based on activation of specific proteinases in plants genus Triticum during fungal infection. Based on these experimental data and genome analysis, the authors are trying to link proteolytic activity they have found in proteomics experiments with a certain family(s) of proteases from a plant genome. This approach is potentially very powerful and might bring a potential leads for the future study.

I only see some minor issues in the experimental approach, which I would like to address:

1. In the experimental setup, where a host is under attack of a pathogen, two organisms are involved. Therefore all the analysis should be done with both Titicum genome/proteome and its pathogens. They should be included in one search space to have one statistical model.

2. Statistical analysis for a database search (f.e. FDR) and parameters for the search are not described as well as the criteria for protein ID filtering (f.e. number of peptides, score, etc) 

3. Please carefully check with a MS specialist the chapter about LC-MS/MS experiment, especially citing of FASP and description of details for LC-MS experimental setup.

4. Please check again the manuscript with a spell-checker in order to avoid non-crucial, but annoying typos like in line 422 should be KMnO4

Author Response

1.  We reperformed all the analyses using the database containing Triticum aestivum and corresponding pathogens genomes. Indeed, it slightly changed the results and made them more specific.

2. The whole procedure for search and protein identification was more clearly and fully described in Materials and Methods section.

3. Appropriate changes were introduced into the text.

4. The text was carefully checked.

Reviewer 2 Report

The  authors have used LC-MS/MS to characterize the degradome of wheat. The application is interested and the mass spectrometry together with bioinformatic tools have been used.

1. In places the article is nicely written and in others it is not. One author has chosen to avoid normal sentence structure  by using colons (:) wherever possible. In places this works OK to introduce lists , although in others it seems as if there was no effort to write a sentence.

2. In terms of science I was suprised how the results suddenly opened with "Release 39 proteome", as I was not expecting a bioinformatic derive opening, nor such a technical attribute.

The use of the phylogenic trees and so on should have been more clearly introduced/pre-warned in the abstract and introduction.

3. For the mass spectrometry part the instrument parameters are mosly well described, although with some rather confusing Obitrap Fusion details that seemed odd.

4. However, the analysis of the proteomic data is scarcely described at all. There is no indication of false discovery rates, peptide scores or indication of what or how anything was identified at all.

My suggestions are as follows:

Indicate that this is a combination of bioinformatics and proteomics in a clearer fashion.

(how well are the hmmer and pfam tools and operations known to the readers in general)

Clarification of the proteomics data analysis MUST be made, indicating how confident the data was for detected peptides or whatever. A flow chart could help.

5. Improve the style of the language, for example:

Why did "Genome A was obtained from Triticum urartu, Genome B from an unknown grass related
to Aegilops speltoides and Genome D from Aegilops tauschii." suddenly occur in the opening?

In the attached document several areas are highlighted where some attention is needed. Apart from maverick punctuation (which is not always so bad), sentences that do not make sense are not useful.

Author Response

1. All places in the text containing colons were checked and rewritten.

2. To address this concern, we added the following passage in the introduction section

An LC-MS approach has been adopted in this study for characterization of all wheat proteases that were identified using Pfam identificators attributed to all the proteases with the use of profile hidden Markov model (HMMER) algorithm [41] that was previously successfully used for identification of proteins [42]. We performed classification of identified proteases.

Also this was shortly pre-warned in the abstract.

3. This missing information was added.

4. LC-MS/MS analysis part was rewritten and missing information was added in Materials and Methods section:

Raw data were processed using IdentiPy search algorithm [48]. Peptide scoring was based on renormalized Hyperscore from X!Tandem [71] (RNHS), where the sum of intensities of the matched peaks was normalized by the total intensity. RNHS is directly used for PSM ranking. Postsearch analysis and FDR filtering relied on MP score algorithm [72]. The MP score is a measure of PSM quality orthogonal to the search score, calculated using a number of descriptors such as precursor ion mass error, median fragment ion mass error, retention time prediction error, etc.

IdentiPy algorithm suggests autotune feature that allow to reprocess spectra and to optimize initial search parameters. Initial values for parameters were set by default -100:+100 ppm for precursor mass error, 500 ppm for fragment mass error and 5 – for allowed miscleavages. In our study, we used 3 biological and 3 technical replicates for the following samples: healthy Khakasskaya plants, infected by P. recondita Khakasskaya plants at 24 hpi, healthy Darya plants, infected by S. nodorum Darya plants at 24 hpi. The results of preliminary search were filtered to 1% FDR using the target-decoy approach and analyzed statistically to derive the optimal parameters that were adjusted by the algorithm for all searches, performed in this study, within -20:+20 ppm range for precursor mass error, 10-11 ppm for fragment mass error and 1-2 – for allowed miscleavages.

In order to reveal differences in number of proteases in cultivars and between healthy and infected plants, full-specific (full-tryptic and full-AspN) peptides were identified in sequences from the database containing all 1544 wheat proteases. Proteases present in all biological and all technical replicates for each type of sample were taken for further analysis.

In order to identify the sites of in vivo hydrolysis in all proteins from the samples, the search of semi-specific (semi-tryptic and semi-AspN) peptides in the sequences from the database containing whole wheat proteome was conducted. Semi-specific peptides present in all biological and all technical replicates for each type of sample were taken for further analysis.

In order to identify the sites of in vivo processing of proteases, the search of semi-specific peptides within the areas that contained 40 amino acid residues, extracted before and after the first peptidase domain amino acid site, was conducted. For this search, the database containing all 1544 wheat proteases was used. In the case of metacaspases, known sites [32] were used for the search. Peptidases containing semi-specific peptides (100% identical, 100% covered), released after digestion by AspN or Trypsin, as defined by IdentiPy, were identified using the BlastP algorithm [73].

5. Thank you for your revisions in the text. They were carefully considered.

Reviewer 3 Report

The manuscript by Balakireva et al, describes a really well done work. Proteolytic events occurring in plant tissues following biotic stresses is a hot topic. The results are extremely interesting and may be of help for the interpretation of other more general works, even for plants other than wheat or cereals. 

The experimental design is appropriate for the aims, well organized and realized, using modern proteomic approaches.

The discussion sets place the data with efficacy. 

The text is clear.

Author Response

We are greatly appreciate for a high evaluation of our work.

Round  2

Reviewer 2 Report

1. The description of the MS data acqusition parameters appear to be innacurate.

You refer to :

"Panoramic scanning was performed in the mass range from 400 m/s to 1,200 m/s, whilst the tandem scanning of fragment ions from the lower boundary of 110 m/s to the upper boundary was determined by the wire state of the precursor ion, but no more than 2,100 m/s. "

I assume you refer to mass to charge ratios (m/z). What do you mean by the wire state.

You also indicate a parameter for synchronous isolation in the MS2 mode. Did you use this, if so why and how were the data used.

You indicate that Roman Zubrev read the manuscript,  maybe he could recheck your instrument description

2. I am very unsure how the contribution of mass spectrometry was used here.

You claim by name to have identified certain peptidases but there is no indication of by how many peptides or sequence coverage.

There is no indication of what peptides were detected, what was their scores and especially what were the false discovery rates.

3. For a reader familiar with proteomics  and peptide analysis there is a clear absence of detail of what was detected, if anything.

Author Response

1. The description of parameters was revised in the previous version of the text, after first round of review (m/s was replaced with m/z). However, the other missed or inaccurate information was added or revised in the following paragraph (please see lines 476-497 in the manuscript):

“One microgram of peptides in a volume of 1-4 µl was loaded onto the Acclaim µ-Precolumn (0.5 mm х 3 mm, 5 µm particle size, Thermo Scientific) at a flow rate of 10 µL/min for 4 min in an isocratic mode of Mobile Phase C (2% acetonitrile, 0.1% formic acid). Then the peptides were separated with high-performance liquid chromatography (HPLC, Ultimate 3000 Nano LC System, Thermo Scientific, Rockwell, IL, USA) in a 15-cm long C18 column (Acclaim® PepMap™ RSLC inner diameter of 75 μm, Thermo Fisher Scientific, Rockwell, IL, USA). The peptides were eluted with a gradient of buffer B (80% acetonitrile, 0.1% formic acid) at a flow rate of 0.3 μL/min. Total run time including initial 4 min of column equilibration to buffer A (0.1% formic acid), then gradient from 5–35% buffer B over 65 min, 6 min to reach 99% buffer B, flushing 10 min with 99% buffer B and 5 min re-equilibration to buffer A amounted 90 min.

MS analysis was performed at least in triplicate with a Q Exactive HF mass spectrometer (Q Exactive HF Hybrid Quadrupole-OrbitrapTM Mass spectrometer, Thermo Fisher Scientific, Rockwell, IL, USA). The temperature of capillary was 240°C and the voltage at the emitter was 2.1 kV. Mass spectra were acquired at a resolution of 120,000 (MS) in a range of 300−1500 m/z. Tandem mass spectra of fragment were acquired at a resolution of 15,000 (MS/MS) in the range from 100 m/z to m/z value determined by a charge state of the precursor, but no more than 2000 m/z. The maximum integration time was 50 ms and 110 ms for precursor and fragment ions, respectively. AGC target for precursor and fragment ions were set to 1*106 and 2*105, respectively. An isolation intensity threshold of 50,000 counts was determined for precursor’s selection, and up to top 20 precursors were chosen for fragmentation with high-energy collisional dissociation (HCD) at 29 NCE. Precursors with a charged state of +1 and more than +5 were rejected and all measured precursors were dynamically excluded from triggering of a subsequent MS/MS for 20 s.”

2. The supplemental material (Tables S1-S12) was added with missed information for each identified protein such as peptides, their coverage, and their scores calculated in every replicate. False discovery rates were also added in the description of protein identification (please see lines 532-534 in the manuscript):

“Peptide-Spectrum Matches (PSMs), peptides and proteins were validated at a 1.0% FDR estimated using the decoy hit distribution. Only proteins having at least two unique peptides were considered as positively identified.”

3. Supplementary material now contains more full information on protein identification. The contribution of MS in the study was carefully revised (please see lines 535-550 in the manuscript):

“In order to reveal differences in number of proteases in cultivars and between healthy and infected plants, full-specific (full-tryptic and full-AspN) peptides were identified in sequences from the database containing all 1544 wheat proteases (Tables S1-S4). Proteases that were covered by at least two unique peptides that were present in all biological and all technical replicates for each type of samples were taken for further analysis.

In order to identify the sites of in vivo hydrolysis in all proteins from the samples, the search of semi-specific (semi-tryptic and semi-AspN) peptides in the sequences from the database containing whole wheat proteome was conducted (Tables S5-S8). Semi-specific peptides present in all biological and all technical replicates for each type of sample were taken for further analysis.

In order to identify the sites of in vivo processing of proteases, the search of semi-specific peptides within the areas that contained 40 amino acid residues, extracted before and after the first peptidase domain amino acid site, was conducted (Tables S9-S12). For this search, the database containing wheat proteases that were identified with the use of full-specific peptides (Tables S1-S4) were used. In the case of metacaspases, known sites [32] were used for the search. Peptidases containing semi-specific peptides (100% identical, 100% covered), released after digestion by AspN or Trypsin, as defined by IdentiPy, were identified using the BlastP algorithm [75].”

Round  3

Reviewer 2 Report

The manuscript now includes sufficient details of the methodology and is suitable for publication.

There may be a few small punctuation errors: E.g.

In total, 1,544 proteases were found, encoded in the whole genome of T. aestivum

Should drop a comma and be  "were found to be incoded"